# Trends in Prescription Chronic Pain Medication Use before and during the First Wave of the COVID-19 Pandemic in Québec, Canada: An Interrupted Time Series Analysis

**DOI:** 10.3390/ijerph20156493

**Published:** 2023-08-01

**Authors:** Gwenaëlle De Clifford-Faugère, Hermine Lore Nguena Nguefack, Manon Choinière, M. Gabrielle Pagé, Lucie Blais, Line Guénette, Marc Dorais, Anaïs Lacasse

**Affiliations:** 1Department of Health Sciences, Université du Québec en Abitibi-Témiscamingue, Rouyn-Noranda, QC J9X 5E4, Canada; gwenaelle.declifford-faugere@uqat.ca (G.D.C.-F.); herminelore.nguenanguefack@uqat.ca (H.L.N.N.); 2Research Center of the Centre Hospitalier de l’Université de Montréal, Montreal, QC H2X 0A9, Canada; manon.choiniere@umontreal.ca (M.C.); gabrielle.page@umontreal.ca (M.G.P.); 3Department of Anesthesiology and Pain Medicine, Université de Montréal, Montreal, QC H3T 1J4, Canada; 4Faculty of Pharmacy, Université de Montréal, Montreal, QC H3T 1J4, Canada; lucie.blais@umontreal.ca; 5Faculty of Pharmacy, Université Laval, Québec, QC G1V 0A6, Canada; line.guenette@pha.ulaval.ca; 6Axe Santé des Populations et Pratiques Optimales en Santé, Centre de Recherche du CHU de Québec—Université Laval, Québec, QC G1V 4G2, Canada; 7StatSciences Inc., Notre-Dame-de-l’Île-Perrot, QC J7V 0S2, Canada; marc.dorais.statsciences@gmail.com

**Keywords:** pain management, opioids, analgesics, nonsteroidal anti-inflammatory drugs, cannabinoids, drug claims

## Abstract

Background: In Canada, a state of health emergency was declared in May 2020 as a result of the COVID-19 pandemic. This study aimed to assess trends in the use of prescription medication for pain management by people living with chronic pain before and during the first wave of the pandemic. Methods: Participants (n = 177) were adults reporting chronic pain who had completed a web-based questionnaire in 2019 and for whom complete longitudinal private and public insurance prescription claims were available. The monthly prevalence of medication use for nonsteroidal anti-inflammatory drugs (NSAIDs), opioids, and prescribed cannabinoids was assessed. An interrupted time series analysis was then performed to evaluate if the COVID-19 pandemic had had an impact on trends in pain medication use. Results: The beginning of the first wave of the pandemic was associated with the onset of a downward trend in opioid use (*p* < 0.05); no such association was found regarding NSAIDs. However, point prevalence of opioid use at the beginning (Nov. 2019) and at the end (Mai 2020) of the study period remained somewhat stable (17.0% vs. 16.4%). Regarding prescribed cannabinoids, a gradual increase in use was observed over the entire study period independently from the impact of the first wave of the pandemic (15.3% vs. 22.6%, *p* < 0.05). Conclusion: While the occurrence of the first wave did have an impact on opioid use among people living with chronic pain, access to and use of opioids appear to have returned to normal before the end of the first wave of COVID-19.

## 1. Introduction

Chronic pain (CP) is defined as pain persisting or recurring beyond three months [1]. Despite challenges leading to underdiagnosis, data shows that CP affects over 20% of the population [2]. In the province of Quebec, Canada, alone, this represents over one million people [3]. There are various causes of CP, including disease, accidents/injuries, surgery, and other unknown causes [1]. Moreover, common CP conditions include back pain, fibromyalgia, migraines, neuropathic pain, osteoarthritis, and rheumatoid arthritis. CP has serious impacts on the physical functioning, emotional well-being and quality of life of those afflicted [4,5,6,7,8,9]. It also constitutes a significant economic burden for the patient and the healthcare system or third-party payers [10]. In Canada, the economic burden of CP in terms of direct healthcare costs and loss of productivity is CAD 38.3–40.4 million per year [11]. Moreover, it is estimated that CP costs more than heart diseases, diabetes or cancer [12]. Despite decades of research on CP and its treatment, the management of this condition remains suboptimal. Indeed, CP is characterized as poorly recognized, underdiagnosed, and frequently inadequately treated [13,14,15,16,17]. This depiction was recently reiterated by Health Canada’s Canadian Pain Task Force [11,18].

Optimal management of CP requires a multimodal approach combining pharmacological, physical, and psychological courses of action [19]. Pharmacological treatments may include different medication classes, such as nonsteroidal anti-inflammatory drugs (NSAIDs), opioids, and cannabinoids. Physical and psychological treatment approaches often involve the intervention of health care professionals (physiotherapists, psychologists, acupuncturists, chiropractors, etc.) or the implementation of self-management strategies (exercise, meditation, etc.). Although multimodal treatment is required, a careful balance between pharmacological and physical/psychological approaches can be hard to achieve and easily disrupted [20,21], e.g., by life events or crises.

In Canada, the state of health emergency was declared in March 2020 due to community transmission of COVID-19. This resulted in the temporary closing of schools, workplaces, and non-essential services [22]. People living with chronic diseases were particularly impacted by these changes [23]. Indeed, recent research results have highlighted a substantial decrease in or interruption of CP-related physical and psychological treatments due to the COVID-19 pandemic [24]. Considering this drop in the use of non-pharmacological approaches, medication use may also have been impacted during the pandemic. In fact, added stress, fear of attending medical appointments, financial strain, and the public health restrictions imposed to cope with the pandemic may have affected access to medication [25]. In addition, in the province of Quebec, medication shortages (such as opioid shortages) [24,26] may have acted as a contributing factor.

During the COVID-19 pandemic, modifications in pharmacological treatments were reported by people living with CP [24,27,28]. Survey data suggests that persons living with CP experienced an increase in analgesics use [24,27], especially for opioids and cannabis [28]. One Colombian study using prescription claims assessed the continuity of analgesic use for CP and revealed an interruption of pain medication use during the pandemic [29]. Another administrative data study conducted in the US found a significant decrease in NSAID use for low back pain, but no change for opioid use [30]. However, in Canada, an increase in opioid use was highlighted by self-reported studies [24,28]. To our knowledge, no Canadian study has yet assessed changes in medication use using administrative databases (such as prescription claims). This type of study could be especially relevant to complete survey data studies. Indeed, medico-administrative databases provide added value through objective longitudinal measures whereby recall bias does not affect results [31]. Also, in Quebec and throughout Canada, prescription claims are well recognized for their validity [32]. To date, no Canadian study has assessed the impact of the pandemic on medication use for chronic pain using medico-administrative databases. As the most reliable approach for measuring medication use trends over time, the analysis of prescription claims allows for the application of interrupted time series analysis using aggregated data collected at regular intervals both before and after an intervention/event. The key assumption is that the data trends observed prior to the intervention can be extended to predict the trends that would have occurred if the intervention/event had not taken place. Studying variations in medication usage at the population level during the pandemic is important because it can help determine if, despite all the disruptions caused by the crisis, effective measures were quickly implemented and allowed patients to avoid experiencing gaps in their treatment.

As such, our study aimed to assess trends in the use of prescription medication for pain management (nonsteroidal anti-inflammatory drugs, opioids, and cannabinoids) among people living with CP before and during the first wave of the COVID-19 pandemic. Specific objectives were: (1) To describe the monthly prevalence of non-steroidal anti-inflammatory drugs (NSAIDs), opioids, and prescribed cannabinoids use before and during the first wave of the COVID-19 pandemic, and (2) to apply an interrupted time series analysis to assess if the pandemic had an impact on trends in medication use.

## 2. Materials and Methods

### 2.1. Study Design and Data Source

In the province of Quebec, the first wave of the COVID-19 pandemic started in March 2020. Daily case numbers peaked in April and decreased in July 2020 [22]. Data from the ChrOnic Pain treatment (COPE) Cohort was used in the present study [33]. The COPE Cohort is comprised of 1935 French-speaking adults who self-reported living with CP (persisting or recurring pain for more than three months) [1]. These participants were recruited in the province of Quebec in 2019 and permission was requested to link their data reported in a web-based questionnaire to longitudinal Quebec administrative databases (private [34] and public [35] prescription claims). The present secondary analysis was conducted among the convenience sample of 177 adults for whom complete longitudinal prescription claims were available to the research team [34] from November 2019 to May 2020 (four months prior to and three months following the onset of the first wave of the pandemic). Our convenience sample was justified by the availability of private prescription claims to the research team at the time of analysis (public prescription claims were not available yet). The sample of 177 adults was clinically comparable to the entire COPE Cohort in terms of mean age and proportion of persons identifying as women, employed, having completed a post-secondary education, reporting pain for ≥10 years, or reporting moderate-to-severe pain [33].

#### Quebec Context and Administrative Databases

Quebec is the only French-speaking province among the ten provinces and three territories of Canada, a country with a universal healthcare system where all essential healthcare services are covered. The Régie de l’assurance maladie du Québec administers provincial public health and drug insurance [36]. This health insurance covers the cost of medical visits, emergency department visits, hospitalizations, and procedures for all Quebec residents (eight million people) [37]. Regarding prescription drugs, only a portion of the population is covered by the Régie de l’assurance maladie du Québec: (1) those not eligible for private drug insurance with their employer or their spouse’s employer; (2) those aged 65 or older; and (3) those receiving last-resort financial assistance. Among the general population, approximately 45% of people fit into one of these three categories [38]. However, when it comes to persons living with CP, 68% are eligible for the Régie’s prescription drug coverage [32]. The public prescription claims contain detailed information on the dispensing of prescription drugs covered by the public plan, and the validity of their contents has been demonstrated [34]. As prescription drug insurance in Quebec is mandatory, the rest is privately covered. In the province of Quebec, researchers can work with the reMed plateform [39]. reMed is an infrastructure that allows the collection of information on the dispensing of prescription drugs for individuals covered by private insurance plans. This registry, established in 2009, contains various essential variables for conducting different pharmacoepidemiological studies, such as the date of purchase, the name, dose, form, and quantity of the prescribed medication, the duration of the prescription, the dosage, the identification number of the pharmacy, the identification number of the prescriber, and the specialty of the prescriber. reMed now includes over 43,700 participants from different regions of Quebec but can also be used in a “project” mode, as in the case of the present study, meaning that consent can be requested from a specific group of participants to obtain their insurance data.

### 2.2. Study Variables

Three classes of pain medication were analyzed and identified using Anatomical Therapeutic Chemical (ATC) codes. In the ATC classification system, active substances are divided into different groups according to the organ or system on which they act (1st level), their therapeutic subgroup (2nd level), and their pharmacological/chemical subgroup (3rd to 5th levels) [39]. For pain management, different classes of medication can be used, including different analgesics and co-analgesics (e.g., nonsteroidal anti-inflammatory drugs, opioids, cannabinoids, antidepressants, anticonvulsants, muscle relaxants, topical anesthetics, etc.) [40,41].

Three classes of interest were included: (1) nonsteroidal anti-inflammatory drugs (ATC codes: NSAIDs; M01A); (2) opioids (ATC code: N02A); and (3) cannabinoids (ATC codes: N02BG10, N03AX24, A04AD10, A04AD11). While nabilone is the only synthetic prescription cannabinoid covered by prescription drug insurance in Quebec (A04AD11), all ATC codes attributable to cannabinoids have been listed above to maximize the reproducibility of our methodology. These three groups of medications were chosen as changes in their use were suggested in the literature [24,26,28,30]. Although antidepressants and anticonvulsants can be used for the treatment of CP, those medications are not pain-specific and the indications for which they are prescribed are difficult to isolate using prescription claims. Therefore, they were not considered in our study.

The COPE Cohort questionnaire included items such as age, gender identity, employment, education level, pain duration, pain intensity on average in the past 7 days (0–10 numerical rating scale), pain frequency (continuous vs. occasional), multisite pain (≥2 sites), generalized pain, current use of prescribed pain medications, and current use of over-the-counter pain medications.

### 2.3. Statistical Analysis

The characteristics of our study sample were described using means, standard deviations (SD), and ranges for continuous variables, and frequencies (n) and proportions (%) for categorical variables. For each participant, medication use (yes/no) was assessed for each month and each class of medication from November 2019 to May 2020. Monthly prevalence (0–100) was calculated by dividing the number of users by the total number of followed participants. An interrupted time series analysis [42] was then performed to evaluate if the pandemic had an impact on trends in medication use (identification of data pattern changes between the first wave and the months prior). As stated, this analysis uses aggregated data collected at regular intervals both before and after an intervention/event. The key assumption is that the data trends observed prior to the intervention can be extended to predict the trends that would have occurred if the intervention/event had not taken place. Outside our main analysis, we also verified whether the prevalence of medication use changed over the entire study period (overall, regardless of the pandemic). Cochran’s Q tests [43] were thus applied. This test is similar to the Repeated measures ANOVA procedure (for binary rather than continuous variables) or can be considered as an extension of McNemar’s test for three or more groups. *p*-values < 0.05 were considered statistically significant. All analyses were conducted using SAS^®^ version 9.4 (SAS Institute, Cary, NC, USA).

## 3. Results

### 3.1. Participant Characteristics

Participant characteristics are presented in Table 1. Among the 177 participants, 83% identified as women and the average age was 50.25 ± 12.36 years (range: 19–77). When looking at self-reported information collected through the web-based questionnaire, 88% of participants reported using prescription pain medications and 63% reported using over-the-counter pain medications. Most participants experienced pain continuously (84%) and half had had pain for more than 10 years. Most participants (85%) also reported multisite pain. Up to 63% were not employed at the time of the survey.

### 3.2. Impact of the Pandemic on Trends in Medication Use

Figure 1 illustrates the trends in pain medication use (prevalence) before and during the first wave of the COVID-19 pandemic (as measured using prescription claims). The pre-pandemic period is represented in the white area of the Figure and the first wave of the pandemic period in the grey area. In the pre-pandemic period, NSAIDs were used by 16.4% of participants, opioids by 17.0%, and prescribed cannabinoids by 15.3% (non-mutually exclusive groups). In May 2020, NSAIDs were used by 17.0% of participants, opioids by 16.4%, and cannabinoids by 22.6%. Although point prevalence of medication use at the beginning and at the end of the study period was similar, interrupted time series analyses revealed that the beginning of the COVID-19 pandemic (March 2020) had a significant impact on opioid use trends (drop in use; time series *p* < 0.05, statistically significant). No statistically significant trend changes before and after the beginning of the pandemic were found for NSAIDs or cannabinoids (*p* > 0.05).

### 3.3. Other Trends in Medication Use Not Related to the Pandemic

We also verified if the prevalence of medication use changed over the entire study period (regardless of the pandemic). While the interrupted time series analysis showed that trends in prescribed cannabinoid use were not affected by the onset of the first wave, a gradual increase in the prevalence of prescribed cannabinoid use was visible starting before the beginning of the pandemic (15.3% in November 2019 vs. 22.6% in May 2020; Cochran’s Q test *p* = 0.0059, statistically significant). No such overall trends were identified for opioids and NSAIDs.

## 4. Discussion

This study aimed to harness the strengths of longitudinal prescription claims to assess trends in prescription medications used by people living with CP before and during the first wave of the COVID-19 pandemic. With such evidence, we can assess how the healthcare practices and medication patterns for individuals with CP may have been affected. This information can help identify any shifts in medication use, potential disruptions in access to medications, and the overall impact of the pandemic on the management of CP, the goal being to inform healthcare strategies. While the beginning of the first wave of the pandemic had a significant impact on opioid use, utilization quickly returned to normal before the end of the first wave. Also, a significant increase in prescribed cannabinoid use—starting before the pandemic—was identified.

### 4.1. Access to Prescribed Medication during the COVID-19 Pandemic

Comparing trends in medication usage during the pandemic across different regions and countries can provide insights into how various contexts have allowed patients to maintain their pharmacotherapy without constraints. Recent epidemiologic studies about CP treatment conducted in the US [30] and in Colombia [29] found the COVID-19 pandemic had an impact on pain medication use. Unlike the results of the present study, the US study revealed a significant decrease in NSAID use, but no change for opioid use [30]. The Colombian study [29] found an interruption in pain medication use during the COVID-19 pandemic. Non-pain-specific US and European claim studies [44,45] showed a decrease in prescription medication use for chronic diseases during the pandemic. Krulichova et al. [45] also highlighted supply issues for drugs used to treat chronic diseases. The observed downward trend in medication use in these various studies may be related to many factors. For example, shortages of opioids were announced during the first wave of the pandemic [46]. Furthermore, many primary care and multidisciplinary pain treatment clinics were closed or reduced their services [25,28]. Fear of going to healthcare appointments [25] was also experienced. In contrast to the findings of the above-mentioned studies, Quebec appears to have experienced a short-lived decrease in medication use (only for opioids). This result suggests that effective measures were quickly implemented, including pharmacy delivery services, telemedicine services, prescription extensions, and controlled drug prescription adjustments (form and/or dose). This is also in concordance with Quebec patients reports suggesting that, despite experiencing longer wait times and more difficulties than usual, they had access to prescribers during the pandemic [26].

### 4.2. NSAIDs

As stated, a US study revealed a significant decrease in NSAID use during the pandemic [30]. We were surprised not to detect any impact of the onset of the pandemic on the use of this class of medications. In fact, we had suspected a decrease considering the uncertainty regarding the use of NSAIDs during the pandemic [47,48]. Our study thus suggests that access to NSAIDs and patients’ pharmacotherapy in the province of Quebec have not been compromised.

### 4.3. Opioids

According to the time series analysis, a significant decrease in opioid use associated with the onset of the pandemic was observed in our study. However, the decrease in prescription medication use among people living with CP was short-lived in Quebec: three months after the start of the COVID-19 pandemic, trends in opioid use returned to normal. This sudden but temporary decrease is possibly explained by the strict regulations surrounding opioids, which required specific accommodation measures but were quickly implemented. Unlike our results, self-reported Canadian studies among CP populations [24,28] highlighted an increase in opioid use. This discrepancy is possibly explained by the context of these self-reported studies that captured the increase in opioid consumption in terms of dose and frequency among existing users, without necessarily translating into an increase in the general prevalence of opioid use in the population. Previous research has shown that during the COVID-19 pandemic, more adverse effects linked to self-medication were reported, mainly for analgesics, psycholeptics, and antibacterials [49].

### 4.4. Prescribed Cannabinoids

Interestingly, a significant increase was observed in terms of prevalence of prescribed cannabinoid use during the study period, starting before the pandemic. The legalization of recreational cannabis took place in October 2018, 18 months before the pandemic in Quebec. This change may have increased the social acceptability of cannabis in Quebec, which may have translated into an upward trend in prescribed cannabinoid use. In fact, the use of cannabis for CP may be subject to stigma and negative attitudes by professionals [50], which could explain why people are turning to prescribe synthetic cannabinoids such as nabilone. The increase in the number of cannabinoid prescriptions in Quebec is an important finding that should be investigated in future studies, for example, concerning explanatory factors from the clinicians’ perspective (qualitative studies), or quantitative studies regarding the determinants and impacts of this usage. Join point regression could also be used in larger samples and over larger time periods to identify the years when changes in cannabinoids use took place.

### 4.5. Strengths and Limitations

This study harnessed the strengths of prescription claims to study longitudinal patterns in the use of prescription medication. In fact, such data sources contain precise data on a very large number of individuals over long periods of time. They also include no recall bias, high specificity of dates and offer high efficiency [32]. Another strength includes the diversity of our sample in terms of pain characteristics and regions of the province of Quebec.

However, some limitations do exist, including the inability to capture the use of over-the-counter medications or cannabis, or the possibility that purchased medication (reported through claims) was not consumed. Furthermore, we only targeted the three classes of medication specific to chronic pain, and the influence of the COVID-19 pandemic on the use of other medication classes such as antidepressants and anticonvulsants has not been investigated. In addition, our prevalence measures were calculated at the population level and do not capture whether it is the same individuals who were using the medications and contributing to the numerator. In fact, if a person stopped opioids and another started such treatment, our study estimating the general prevalence would not capture it. While our sample was sufficient for our descriptive statistics and time series analysis [51], the follow-up period was short and the sample size did not allow for multivariate analyses to identify factors predicting change in individual drug use. In terms of external validity, our sample is reflective of the COPE Cohort (over-representation of women, but comparable to random samples of Canadians living with CP in terms of age, employment, post-secondary education, pain duration, and pain intensity) [33].

## 5. Conclusions

The objective of this study was to utilize the advantages of longitudinal prescription claims to analyze the patterns of prescription medication usage among individuals living with CP prior to and during the first wave of the COVID-19 pandemic. While the occurrence of the first wave did have an impact on trends in use of certain types of medications used by people living with CP such as opioids, the present study has shown that access and use appear to have returned to normal before the end of the first wave. Effective measures were quickly implemented and allowed patients to avoid experiencing gaps in their treatment. It will therefore be possible to draw inspiration from these Quebec measures during future health crises. In addition, a gradual and marked increase in the use of prescribed cannabinoids (nabilone) was detected over the study’s seven-month period. This is probably due to the increasing acceptability of cannabinoids following cannabis legalization in Canada (2018).

## Figures and Tables

**Figure 1 ijerph-20-06493-f001:**
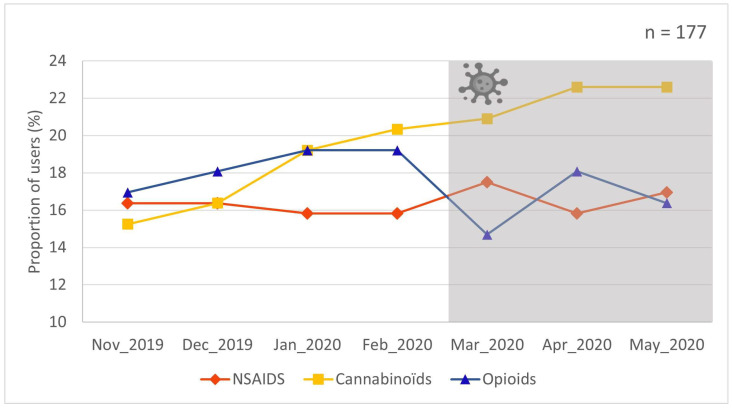
Trends in pain medication use before and after the first wave of the COVID-19 pandemic. Interrupted time series analysis revealed data pattern changes between the pre-pandemic period (white area) and first wave of the pandemic period (grey area) for opioids only (*p* < 0.05, statistically significant).

**Table 1 ijerph-20-06493-t001:** Characteristics of the sample.

Characteristics (n = 177)	n (%)
Age (years)—mean ± standard deviation	50.3 ± 12.7
Range	19–77
Gender identity *	
Women	147 (83.0)
Men	30 (17.0)
Employed	
Yes	65 (36.7)
No	111 (62.7)
Missing data	1 (0.6)
Post-secondary education	
Yes	148 (83.6)
No	27 (15.3)
Missing data	2 (1.1)
Pain duration	
<1 year	2 (1.1)
1–4 years	41 (23.2)
5–9 years	44 (24.9)
≥10 years	90 (50.8)
Pain intensity (0–10 NRS) on average in the past 7 days	
Mild (1–4)	64 (36.6)
Moderate (5–7)	91 (52.0)
Severe (8–10)	20 (11.4)
Pain frequency	
Continuous pain	148 (83.6)
Occasional pain	28 (15.8)
Missing data	1 (0.6)
Multisite pain (≥2 sites)	
Yes	151 (85.3)
No	26 (14.7)
Generalized pain	
Yes	54 (30.5)
No	123 (69.5)
Current use of prescribed pain medications	
Yes	155 (87.6)
No	21 (11.8)
Missing data	1 (0.6)
Current use of over-the-counter pain medications	
Yes	111 (62.7)
No	65 (36.7)
Missing data	1 (0.6)

Notes: * Non-binary was also an option; however, none of the 177 individuals self-identified as such. Abbreviations: NRS: Numeric rating scale.

## Data Availability

The COPE Cohort dataset is not readily available because participants did not initially provide consent to open data. The data that support the findings of this study are available from the corresponding author upon reasonable request and conditionally to a proper ethical approval for a secondary data analysis. Programming codes can be obtained directly from the corresponding author.

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
