# Peer review of "Trends in Prescription Chronic Pain Medication Use before and during the First Wave of the COVID-19 Pandemic in Québec, Canada: An Interrupted Time Series Analysis"

_ijerph, 2023, doi:10.3390/ijerph20156493_

Round 1

Reviewer 1 Report

Attached the comments 

Author Response

Please see the attached document for our point-by-point response to all received comments.

Reviewer 2 Report

Chronic pain is an important health problem nowadays and affects millions of people around the globe. Thus, the topic of this article is particularly interesting and up to date.

There are some problems with this manuscript:

1) The introduction does not explicitly state the research problem or gap that motivates the study

2) In the introduction the authors use some old references (up to 2010) to support the claims about the impact of the COVID-19 pandemic on pharmacological and non-pharmacological approaches to chronic pain management. Reference should be made to more recent studies to gain an up-to-date perspective on the issue.

3) It is not clearly stated what the objectives of the presented study are. Research objectives should be explicitly stated to clarify the direction of the study and highlight its original contribution.

4) The article contains long and complex sentences, which may make understanding more difficult.

5) The authors should provide more information about the exact period of the study, clarify the description of the drug classes analyzed and provide a more detailed explanation of the reasons why some drug classes were excluded (antidepressants, anticonvulsants).

6) The discussion chapter does not adequately address the possible causes of the decline in drug use during the pandemic. Mentioning possible problems with drug supply is not enough; further evidence or discussion of the exact impact of these issues on medication use would be needed.

Author Response

(The authors gave the same response as above.)

Round 2

Reviewer 1 Report

Thank you for the changes 

Author Response

We thank the reviewer for the time and effort invested into the review of our manuscript.

Reviewer 2 Report

The authors took into account the observations made and in this sense they improved the manuscript.

Author Response

(The authors gave the same response as above.)
